# Development and validation of a subjective end-of-life health literacy scale

Clément Meier[1,2,3]*, Sarah Vilpert[1,3], Maud Wieczorek[4], Carmen Borrat-Besson[3], Ralf J. Jox[5], Jürgen Maurer[1]

1 Faculty of Business and Economics (HEC), University of Lausanne, Lausanne, Switzerland, 2 Faculty of Biology and Medicine (FBM), University of Lausanne, Lausanne, Switzerland, 3 Swiss Centre of Expertise in the Social Sciences (FORS), Lausanne, Switzerland, 4 Swiss Centre of Expertise in Life Course Research LIVES, Lausanne and Geneva, Geneva, Switzerland, 5 Palliative and Supportive Care Service, Chair in Geriatric Palliative Care, and Institute of Humanities in Medicine, Lausanne University Hospital and University of Lausanne, Lausanne, Switzerland

* clement.meier@unil.ch

**Data Availability Statement:** Data cannot be shared publicly because they are part of the SHARE project, which has its own data access policies. These data are available from the SHARE project for researchers who meet the criteria for access to

## Abstract

Personal health literacy is the ability of an individual to find, understand, and use information and services to inform health-related decisions and actions for oneself and others. The end of life is commonly characterized by the occurrence of one or several diseases, the use of many different types of healthcare services, and a need to make complex medical decisions that may involve challenging tradeoffs, such as choices between quality and length of life. Although end-of-life care issues concern most people at some point in life, individuals' competencies to deal with those questions have rarely been explored. This study aims to introduce, develop, and validate an instrument to measure individuals' self-assessed competencies to deal with end-of-life medical situations, the Subjective End-Of-Life Health Literacy Scale (S-EOL-HLS), in a sample of older adults aged 50+ living in Switzerland who participated in wave 8 (2019/2020) of the Survey of Health, Ageing, and Retirement in Europe. The S-EOL-HLS uses a series of questions on self-rated difficulties in understanding end-of-life medical jargon, defining in advance which end-of-life medical treatments to receive or refuse, and communicating related choices. Aside from conducting exploratory and confirmatory factor analysis to evaluate the construct validity, we compared measurements from the S-EOL-HLS to respondents' general health literacy measured with the European Health Literacy Survey questionnaire. We obtained a three-factor model with acceptable fit properties (CFI = 0.993, TLI = 0.992, RMSEA = 0.083, SRMR = 0.061) and high reliability (α = 0.93). The partial associations between the health literacy scores from the two scales and respondents' sociodemographic characteristics were similar; however, individuals with higher end-of-life health literacy scores appeared to have more positive attitudes towards end-of-life care planning outcomes. The S-EOL-HLS demonstrates reliable and consistent results, making the instrument suitable for older adults in population surveys.

confidential data. More information about data access and the Conditions of Use can be found on the SHARE project's website (https://share-eric.eu/data/data-access/conditions-of-use).

**Funding:** This study was funded by the Swiss National Science Foundation (SNSF) for the end-of-life project. Healthy Ageing in the Face of Death: Preferences, Communication, Knowledge, and Behaviors Regarding End of Life and End-of-life Planning Among Older Adults in Switzerland (grant number: 10001C_188836) awarded to Prof. JM, CM is employed for this project. The funders had no role in study design, data collection, and analysis, decision to publish, or preparation of the manuscript.

**Competing interests:** The authors declare that they have no competing interests.

## Introduction

Personal health literacy, that is, "the degree to which individuals can find, understand, and use information and services to inform health-related decisions and actions for themselves and others" [1], is commonly seen as a crucial factor in enabling patients' autonomy, improving their satisfaction and achieving better health and healthcare outcomes [2]. By acquiring health literacy skills, individuals can make healthier lifestyle choices, seek more appropriate healthcare services, and empower themselves to deal with illness [3, 4]. Health literacy skills hold particular importance in aging populations where chronic diseases are more prevalent [5]. These skills are essential due to the frequent need for complex treatment regimes, especially in cases of multimorbidity [5]. Health literacy significantly influences how individuals perceive their health challenges, communicate with healthcare providers, and make medical decisions [6].

Health and healthcare decision-making regarding the end of life, such as the completion of advance directives or the engagement in advance care planning, can be especially challenging for individuals as they have to make anticipatory decisions for hypothetical scenarios that may involve complex tradeoffs between quality and quantity of life and that frequently take place in emotionally-charged situations with potentially difficult family dynamics and interactions [7–9]. End-of-life health literacy is likely to be distinct from general health literacy due to the particular skills needed to navigate specific challenges of end-of-life planning and decision-making, such as exceptionally high stakes (questions of life and death), increased levels of risk and uncertainty, major emotional challenges related to the family and social contexts of dying, and complex discussions with highly-specialized professionals, potentially in a context of deteriorating physical and/or mental capacities. Poor end-of-life health literacy can lead to fewer palliative care visits, lack of advance care planning and advance directive completion, worse health status, lower quality of life at the end of life, and higher rates of unnecessary hospitalizations [10–12]. Improving health literacy related to end-of-life decision-making can thus support patient engagement and empowerment to make their own decisions in the face of death and potentially result in improved outcomes related to death and dying [13, 14].

Before delving into the specifics of our study, it's pertinent to shed light on some region-specific factors within Switzerland, the country where our research was conducted. The legal status of advance directives in Switzerland has been firmly established since 2013, when the Swiss Federal Council acknowledged their importance and introduced a new adult protection law into the Swiss Civil Code [15]. This significant legislative step amplified public awareness around end-of-life planning, however, despite this new law and the general growth of interest regarding end-of-life issues and palliative care, research indicates that the proportion of the general population completing advance directives remains relatively low [16]. Furthermore, it is crucial to acknowledge Switzerland's significant linguistic diversity, with the country partitioned into three main regions: German, French, and Italian. Differences across these regions are often observed, notably in the varying preferences, attitudes, and behaviors related to end-of-life care and planning [16, 17].

Even though end-of-life care issues concern every person, little research has explored the level of health competencies that individuals have for this life stage. So far, existing studies have shown that patients' knowledge of end-of-life care options is rather limited [18, 19], which may at least partly reflect limited competencies to deal with end-of-life medical situations. Recent population-based studies have highlighted significant knowledge gaps regarding end-of-life care options as well as considerable variation in the perceptions of medical end-of-life situations [20, 21]. In addition, the ACP Engagement Survey was developed to measure the complex process of advance care planning by asking questions on surrogate decision-making, value, and quality of life, and on communication with medical providers; the results ultimately

showed that engagement in advance care planning remains low among older patients [22]. Other research focusing on so-called "death literacy," a concept related to individuals' skills and knowledge regarding the death system, such as factual knowledge, learning experience, emotional support, hands-on care, or community capacity, suggests that higher death literacy could help individuals make better-informed decisions regarding end-of-life and death care options [23]. While the concept of death literacy encompasses knowledge and skills related to understanding and navigating the death system, it does not specifically address individual's ability to navigate medical decisions at the end of life. Yet, to the best of our knowledge, there is no specific survey instrument to measure end-of-life health literacy in view of the distinct challenges of end-of-life decision-making relative to more general decision-making challenges concerning health and healthcare.

Building on existing international end-of-life research [24] and corresponding evidence from Switzerland [25], our study had three distinct aims.

- First, we aimed to introduce the conceptual basis and the development of a new survey scale—the Subjective End-Of-Life Health Literacy Scale (S-EOL-HLS)—to measure the level of competencies individuals perceive to have in dealing with end-of-life care situations.

- Our second aim was to assess the reliability and construct validity of the new instrument with both exploratory and confirmatory factor analysis in a sample of older adults aged 50+ in Switzerland.

- Finally, our last objective was to compare the respective associations between individuals' social, regional, and health characteristics and the scores from the S-EOL-HLS and from the validated European Health Literacy Survey questionnaire (HLS-EU-Q16). We further checked the discriminant ability of these two instruments with regard to end-of-life care planning outcomes.

## Methods

### Conceptual framework and development of the survey instrument

**Conceptual framework.** The proposed S-EOL-HLS aims to measure individuals' self-perceived end-of-life health literacy skills for advance care planning and end-of-life decision-making. To this end, the instrument aims to measure individuals' subjective ease (1) in comprehending vocabulary that is commonly used in advance care planning and discussions of end-of-life care (functional health literacy); (2) to effectively engage, interact and apply newly-acquired information in discussions with healthcare providers and family concerning advance care planning and end-of-life care (interactive health literacy) and (3) using relevant end-of-life-related information and advice to form and express informed end-of-life decisions that are aligned with the individuals' preferences and values, including potential advance end-of-life decisions as required by advance care planning or when using advance directives (critical health literacy). The conceptual distinction between functional, interactive, and critical health literacy as three key hierarchical layers of general health literacy was first explained in the seminal work of Nutbeam (2000) [26], and was adapted by Ladin et al. (2018) [27] to (advance) end-of-life care decision-making in order to assess health literacy gaps for end-of-life planning among older dialysis patients in the United States. Specifically, Nutbeam's general health literacy framework conceptualizes functional health literacy as individuals' abilities to read and write effectively in everyday situations; interactive health literacy as more advanced skills to participate actively in daily activities to extract and apply information from various forms of communication; and critical health literacy as an even

higher level of skills regarding how individuals critically analyze information to make informed decisions.

**Development of the survey instrument.** Following Nutbeam's framework and using its adaptation to end-of-life health literacy proposed by Ladin et al. (2018) [27], we developed a series of survey items aimed at measuring functional, interactive, and critical health literacy pertaining to end-of-life care decision-making with a focus on critical health literacy skills for the completion of advance directives as key tools of advance care planning in both clinical settings and more broadly, in the general population (S1 Fig).

In general, health literacy can be assessed by subjective or objective (often test-based) measures [28]. These two approaches are often considered complementary, as they capture distinct aspects of health literacy and have each different practical advantages and disadvantages concerning measurement across different settings [28]. Objective test-based measures of health literacy, such as the Test of Functional Health Literacy in Adults (TOFHLA) or the National Assessment of Adult Literacy (NAAL) [29, 30], aim at quantifying individuals' health literacy skills by subjecting them to standardized test stimuli, which measure whether individuals can accurately perform a specific task. While such objective test-based measures of health literacy provide directly comparable measurements of a person's skill in a prespecified domain and do not suffer from issues of differential item functioning, one common limitation of these measures is that they tend to be domain-specific and may, therefore not be easily generalizable to different contexts. In addition, objective test-based measures typically require standardized conditions, thus excluding collaboration with other individuals, which makes such tests easier to administer in-person than, say, in the context of a paper and pencil mail or drop-off survey. Finally, "testing" individuals for their health literacy skills may result in high response burden in terms of required survey time, cognitive effort, and risk for stigma, which makes it challenging to include such assessments in larger-scale longitudinal general-purpose surveys in which considerations of interview time and risk of attrition are often paramount [28]. On the other hand, subjective health literacy measures typically ask individuals to rate their perceived difficulties with different types of health literacy tasks on a Likert or other rating scale. While such measures lack fully-standardized test stimuli and may suffer from differential item functioning, they are also often easier and more rapid to administer, especially in the context of a drop-off survey, less stigmatizing for individuals with lower health literacy, and easily adaptable to a broad range of situations. Subjective measures of health literacy have a notable connection with the concept of self-efficacy in health literacy tasks [31–33]. This connection makes the measurement of subjective health literacy interesting. Specifically, individuals' judgments of their capabilities to execute specific health literacy tasks—i.e., self-efficacy—may be a significant determinant of health behaviors, healthcare use, and related outcomes [34]. Subjective health literacy may thereby be especially important in the context of advance care planning and end-of-life care due to the special challenges posed by such planning and the high importance of individuals' own initiative for engaging in such planning. Subjective health literacy measures such as the European Health Literacy Survey questionnaire (HLS-EU) or the Health Literacy Questionnaire (HLQ) are, therefore, used as important complements to objective test-based measures. Developing subjective health literacy measures specifically for advance care planning and end-of-life decision-making holds major promise to improve our understanding of individuals' engagement in these processes.

Following the approach of other subjective assessments of general health literacy, such as the HLS-EU, we designed several questions (S1 Fig) to assess self-rated/subjective (a) functional end-of-life health literacy (understanding specialized vocabulary items); (b) interactive health literacy (feel comfortable items); (c) critical health literacy (treatment preference items). All items were rated using an identical four-point Likert scale with the answer categories "very

easy," "fairly easy," "fairly difficult," and "very difficult," corresponding to the rating scales used in the HLS-EU and other subjective health literacy assessment tools [35–37]. Specifically, functional end-of-life health literacy (1) is measured by six items assessing self-rated difficulties in *understanding* medical terms that are relevant to end-of-life decision-making (prognosis, intubation, palliative care, cardiopulmonary resuscitation, artificial nutrition, and sedation). Seven items measure interactive end-of-life health literacy; the first three elicit respondents' self-rated difficulties in talking about their end-of-life preferences with someone they trust, such as a close family member or friend; self-rated difficulties in talking to a physician or other medical expert to learn more about advance care planning tools and end-of-life treatments; and self-rated difficulties in finding/obtaining information and/or obtain template forms to complete a so-called "advance directives." The next four items are related to how individuals apply the new information; they measure respondents' self-rated difficulties in defining what "overtreatment" means to them, making decisions on whether to accept a treatment or not based on probabilities of different treatment outcomes, choosing between comfort care (relieving suffering without slowing the disease) and aggressive life-prolonging treatment (heavy chemotherapy, intensive care with artificial ventilation) should they suffer from a terminal disease; and defining specific conditions or situations in which they would prefer to be left to die. Critical end-of-life health literacy is assessed by five items measuring individuals' self-rated difficulties if they had to decide (at the time of the interview) whether they wish to receive or refuse five potential treatments at the end of life (breathing machines, artificial nutrition, blood transfusion, antibiotics, cardiopulmonary resuscitation), which correspond to commonly used "tick box items" in various advance directive forms/templates proposed by different organizations and institutions.

Face and content validity of the S-EOL-HLS was established through an iterative discussion and revision process involving the entire multidisciplinary project team composed of experts in sociology and psychology of health, public health, palliative care, end of life, and survey research. We also conducted six in-depth cognitive interviews with adults aged 50 and over to identify and correct potential confusion or inconsistency in question understanding, lack of clarity, and specificity in question-wording. Finally, our instrument was first distributed to a pilot sample of 123 respondents representing our target population of older adults in Switzerland to analyze potential response biases before being administered to the entire Swiss SHARE sample.

## Validation/psychometric assessment of the instrument

**Study design and participants.** We used data from wave 8 of the Swiss component of the Survey of Health, Ageing, and Retirement in Europe (SHARE) [38]. Every two years, SHARE collects longitudinal information on health, socioeconomic status, and social or family network from Europeans aged 50 years and older and their partners. The study began in 2004 with random representative samples of older individuals in each of the ten participating countries, including Switzerland. SHARE now includes 27 European countries and Israel. During each survey round, respondents give their verbal informed consent to participate in SHARE twice: once when they accept to schedule a personal interview after a phone call from the interviewer and then again at the beginning of the face-to-face interview. The SHARE data combines internationally-harmonized face-to-face Computer-Assisted Personal Interviewing (CAPI) interviews and national self-administered paper-and-pencil questionnaires completed by the respondents after the main in-person interview. Our study includes data from the Swiss self-administered national questionnaire on end-of-life issues, which was issued during SHARE wave 8 (2019/2020) in Switzerland, and sociodemographic variables obtained from

the main SHARE interview. In March 2014, the Ethics Committee of the Canton of Vaud, Switzerland, granted our study the ethical approval, bearing the number 66/14. Overall, 2,005 Swiss respondents participated in Wave 8 of SHARE between October 2019 and the beginning of March 2020 (pre-COVID-19 measures). Among them, 1,891 individuals also completed the Swiss self-administered questionnaire (a 94,3% cooperation rate). After excluding respondents with missing item responses on any variable included in our analysis, our final analytical sample contains 1,270 participants.

**Outcome variables.** *Subjective End-Of-Life Health Literacy Scale (S-EOL-HLS).* The S-EOL-HLS contains 18 items (S1 Table) aimed at measuring functional, interactive, and critical health literacy pertaining to end-of-life care decision-making and skills for completing advance directives. Following the approach used by the HLS-EU consortium, the final end-of-life health literacy score is calculated as the percentage of items that were answered with "very easy" or "easy": (Number of "very easy" or "easy" responses / 18) x 100. The scale thus ranged from 0 to 100.

*HLS-EU-Q16.* The short version of the European Health Literacy Survey questionnaire developed by the HLS-EU consortium was also part of our Swiss paper-and-pencil questionnaire to capture our respondents' self-assessed general health literacy and compare it to the more specific concept of end-of-life health literacy. The HLS-EU scale contains 16 items (S2 Table) with which individuals rate their health literacy with regard to four stages of information processing, i.e., accessing/obtaining health information, understanding health information, processing/appraising health information, applying/using health information, and three domains, i.e., healthcare, disease prevention, health promotion. Each item includes concrete health-relevant tasks or situations whose perceived difficulty respondents rate on a 4-point Likert scale with answers ranging from "very easy," "fairly easy," "fairly difficult," to "very difficult." The final health literacy score is calculated as the percentage of items that were answered with "very easy" or "easy": (Number of "very easy" or "easy" responses / 16) x 100. The scale thus ranged from 0 to 100.

**Independent variables.** Our regression models include information about gender (0 = male, 1 = female), age group (50–64 years, 65–74 years, 75+ years), and education level, which was divided into three categories based on the International Standard Classification of Education (ISCED) of 2017 [39] (low = ISCED levels 0-1-2, middle = ISCED levels 3–4, high = ISCED levels 5–6). Our study looked at all types of partnerships rather than just focusing on legal marriage (0 = has a partner, 1 = has no partner). Based on the question: "Is your household able to make ends meet?" respondents' perceptions of their financial situation were recoded into three categories (1 = easily, 2 = fairly easily, 3 = with difficulty). We also included information on which of the three linguistic regions of Switzerland the respondents lived in, depending on the language they used to answer the questionnaire (German, French, or Italian), as well as if they lived in an urban or rural area (0 = urban, 1 = rural). Finally, we controlled for respondents' self-rated health status and recoded the outer categories to obtain a three-point scale (1 = poor/fair health, 2 = good health, 3 = very good/excellent health).

**End-of-life health outcomes.** Three variables assessing attitudes toward the end of life were also used in the analysis: whether respondents ever discussed with someone about their wishes for the end of their life (1 = yes, 2 = no), whether they have completed a written statement about their wishes and refusals for medical treatments and care (advance directives) (1 = yes, 2 = no) and whether they appointed someone in writing to make medical decisions for them should they not be able to make those decisions for themselves (1 = yes, 2 = no).

**Assessment of metrics properties.** We first investigated the correlation matrix of the 18 proposed end-of-life health literacy items using Pearson correlation analysis and checked the internal consistency and reliability of the items using Cronbach's alpha [40]. Using exploratory

factor analysis on a randomly split-half sample (n = 635), we assessed the unrestricted factor structure of our scale to evaluate the number of factors and their respective dimensions without imposing any of the conceptual designs used during scale development [41, 42]. The exploratory factor analysis was conducted using a weighted least squares estimator and Promax rotation. Following the exploratory factor analysis, we used the second half of the sample (n = 635) for a confirmatory factor analysis to test the presumed three-domain structure of end-of-life health literacy consisting of functional, interactive, and critical end-of-life literacy (S1 Fig) [43]. The confirmatory factor analysis used the weighted least squares mean, and variance adjusted (WLSMV) estimator that best fit the categorical and ordinal nature of the data. We compiled the root mean square error of approximation (RMSEA), standardized root means square residual (SRMR), comparative fit index (CFI), and adjusted goodness of fit index (TLI) to assess the model fit. The following cut-off values were considered as indications of an acceptable fit; RMSEA $\leq$ 0.08, SRMR $\leq$ 0.10, CFI $\geq$ 0.95, and TLI $\geq$ 0.95 [44–46]. Exploratory and confirmatory factor analyses were performed with the software packages *Psych* version 2.2.9 and *Lavaan* version 0.6–12 using R version 4.1.2.

**Assessment of end-of-life subjective health literacy in the older population in Switzerland.** To evaluate the construct validity of the S-EOL-HLS scale, we used OLS regression models to compare the partial associations between the S-EOL-HLS score and respondents' characteristics with those from the HLS-EU-Q16. The estimated standard errors were adjusted to account for the possibility of dependencies in the observations as both partners of the same couple may participate in our study, which increases the chances of similar responses. The regressions were hence clustered at the household level to account for such potential dependencies. In addition, we also compared the two scales' average scores for the three end-of-life care planning outcomes. We finally completed a ROC analysis and used the area under the curve to evaluate the performance of both scales on the three end-of-life planning outcomes [47]. All estimations were performed using STATA/SE 17.0 software (STATA Corporation, College Station, TX).

## Results

Table 1 introduces the key characteristics of the analytical sample. Almost half of the sample were women (52%), the mean age was 70.4 years old (SD: 8.2), and the majority had a middle level of education (64%). More than three quarters of the respondents had a partner (78%), and most reported that it was "easy" (57%) or "fairly easy" (31%) to make ends meet at the end of the month. Regarding the language region, 73% lived in the German-speaking part of Switzerland, and 55% lived in a rural area. Respondents mostly reported good or excellent health (42% and 42%, respectively). Concerning the end-of-life care planning outcomes, 66% had already discussed their wishes for the end of life, 42% had completed an advance directive, and 43% had appointed someone as surrogate to make medical decisions on their behalf.

Fig 1 displays the proportion per category of answers for each end-of-life health literacy item. Most respondents seemed not to have difficulties dealing with end-of-life medical situations. The three medical terms that were most difficult to understand were "sedation" (40.9%), "intubation" (27.3%), and "palliative care" (18.6%). Respondents reported unease in various situations: 39.7% found it difficult to make decisions that involved probabilities; 34.6% struggled to define the term 'overtreatment'; 37.7% had difficulties in specifying conditions or circumstances under which they would prefer to die; and, 36.1% were uncomfortable when asked to choose a type of treatment if they were to have a terminal illness. Finally, it was also rather difficult for respondents to indicate their willingness to receive or refuse "breathing machines"

**Table 1. Characteristics of the study population, adults aged 50+, SHARE Switzerland, 2019/2020, n = 1,270.**

|  | n | % |
|---|---|---|
| **Gender** |  |  |
| Male | 612 | 48 |
| Female | 658 | 52 |
| **Age groups** |  |  |
| 58–64 years | 352 | 28 |
| 65–74 years | 548 | 43 |
| 75+ years | 370 | 29 |
| **Education** |  |  |
| Low | 190 | 15 |
| Middle | 808 | 64 |
| High | 272 | 21 |
| **Partnership status** |  |  |
| Has a partner | 991 | 78 |
| No partner | 279 | 22 |
| **Make ends meet** |  |  |
| Easily | 719 | 57 |
| Fairly easily | 395 | 31 |
| With difficulty | 156 | 12 |
| **Linguistic regions** |  |  |
| German | 920 | 73 |
| French | 308 | 24 |
| Italian | 42 | 3 |
| **Living area** |  |  |
| Urban | 572 | 45 |
| Rural | 698 | 55 |
| **Self-rated health** |  |  |
| Poor/fair health | 202 | 16 |
| Good health | 531 | 42 |
| Excellent health | 537 | 42 |
| **EOL discussion** |  |  |
| Yes | 841 | 66 |
| No | 429 | 34 |
| **Complete ADs** |  |  |
| Yes | 527 | 42 |
| No | 743 | 58 |
| **Appointed surrogate** |  |  |
| Yes | 549 | 43 |
| No | 721 | 57 |

Note, number of observations for the whole sample. AD = Advance Diectives. EOL = End-Of-Life.

(28.6%), "cardiopulmonary resuscitation" (30.3%), and "artificial nutrition" (30.4%) as part of their end-of-life care wishes.

The next heatplot on Fig 2 presents the Pearson correlation analysis of the S-EOL-HLS items. It shows that the items are moderately correlated with each other and show higher correlations among items that aim to measure the same aspect of end-of-life health literacy. The graph identifies the three factors present in the scale: functional end-of-life health literacy

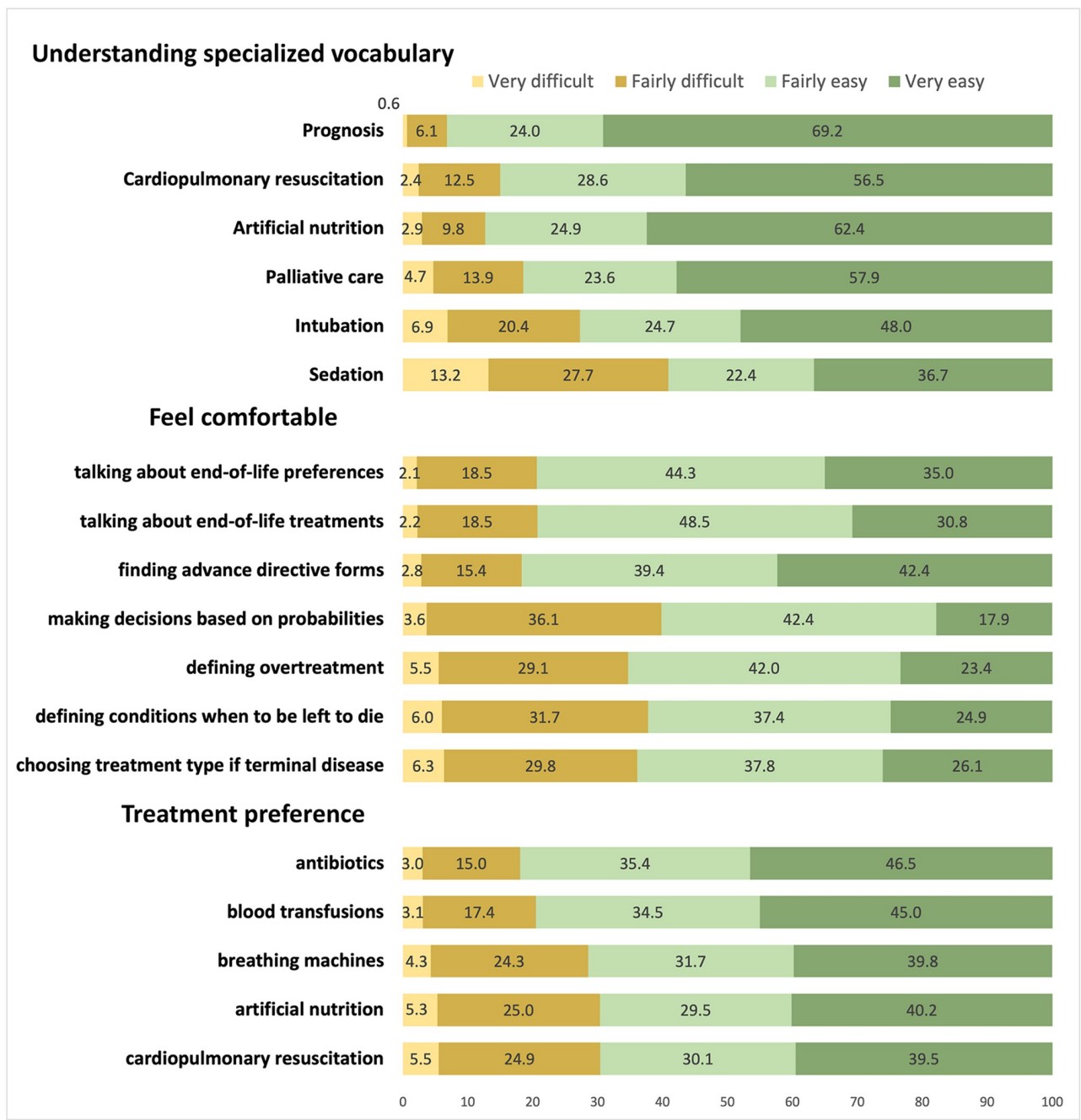

**Fig 1. Subjective End-of-life Health Literacy Scale (S-EOL-HLS), percentage of respondents per categories, adults aged 50+, SHARE Switzerland, 2019/2020, n = 1,270.**

(understanding specialized vocabulary), interactive end-of-life health literacy (feel comfortable), and critical end-of-life health literacy (treatment preference). The Cronbach alphas also indicated high internal consistency and reliability for the overall instrument ($\alpha = 0.93$) as well as for each factor ($\alpha_1 = 0.9$, $\alpha_2 = 0.86$, $\alpha_3 = 0.93$).

We first examined the sampling adequacy and correlation among the items before the exploratory factor analysis. The suitability for performing factor analysis was confirmed with a

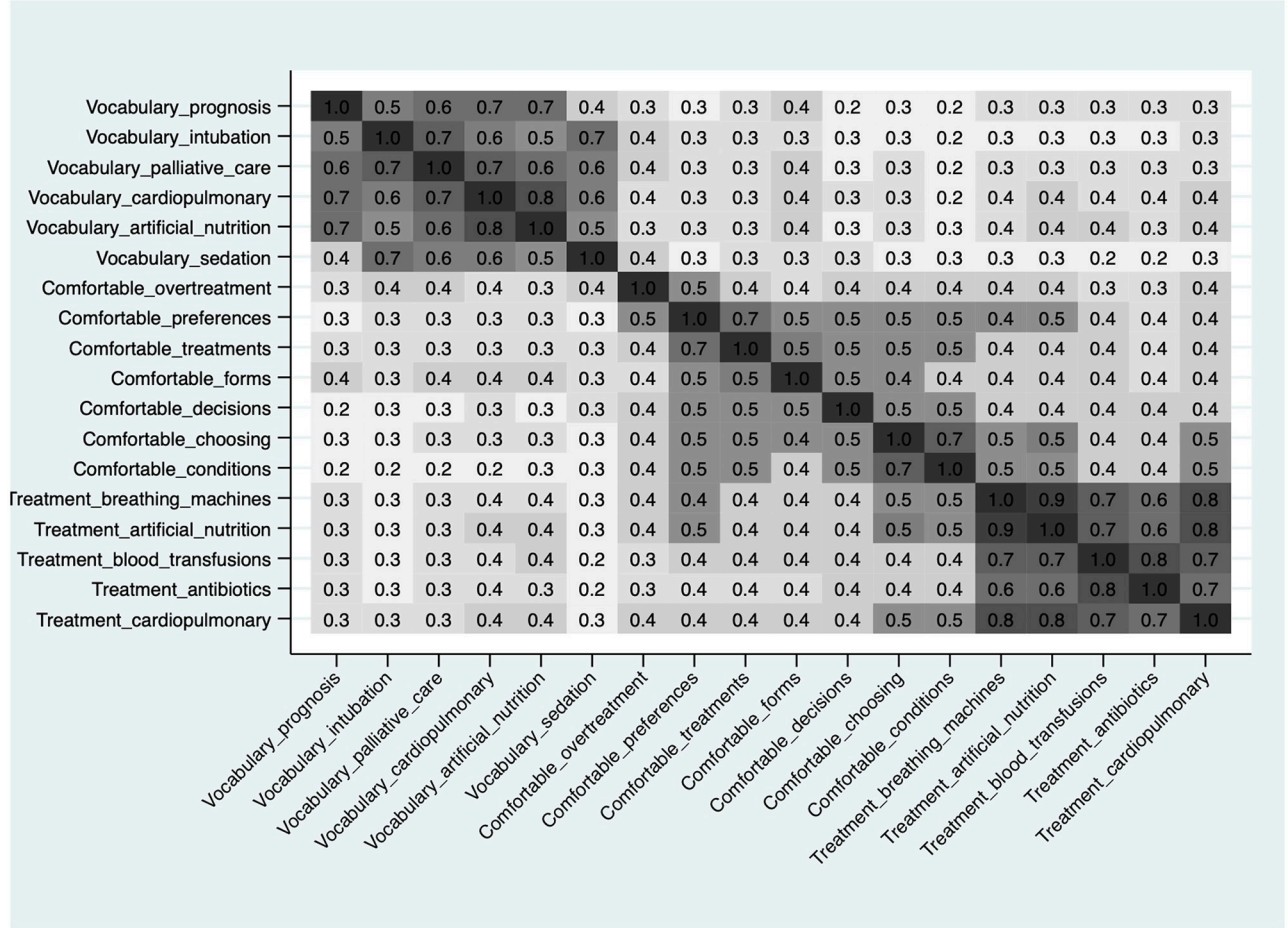

**Fig 2. Heatplot of the S-EOL-HLS items, adults aged 50+, SHARE Switzerland, 2019/2020, n = 1,270.**

Kaiser-Meyer-Olkin (KMO) of 0.92 and a statistically significant Bartlett's test of sphericity (chi square = $X^2$ = 7578.036, df = 153, p-value = 0.000). The initial analysis resulted in a three-factor solution; S2 Fig shows two eigenvalues superior to one, one eigenvalue close to one, and all three within the sharp descent [48]. Table 2 outlines the results of the exploratory factor analysis with Promax rotation. The first factor (functional end-of-life health literacy) included six items that explained 34% of the variance and had rotated factor loadings ranging from 0.66 to 0.87. The second factor (interactive end-of-life health literacy) had seven items that explained 32% of the variance, with rotated factor loadings ranging from 0.48 to 0.80. Finally, the third factor (critical end-of-life health literacy) combined five items that described 35% of the variance and had rotated factor loadings between 0.82 to 0.87. Then, the confirmatory factor analysis validated the three-factor model with acceptable fit properties (CFI = 0.993, TLI = 0.992, RMSEA = 0.083, SRMR = 0.061).

To evaluate the construct validity of the S-EOL-HLS, we compared it to the HLS-EU-Q16. Table 3 shows that the partial associations between the two scales and respondents' sociodemographic characteristics were similar. Women were more likely to have both higher end-of-life health literacy and health literacy scores than men. Also, respondents with a middle or high level of education were more likely to have both higher end-of-life health literacy and health literacy scores. Regarding their financial situation, respondents who stated that they have difficulties making ends meet were less likely than those without difficulties to have high

**Table 2. Results of exploratory factor analysis with Promax rotation, adults aged 50+, SHARE Switzerland, 2019/2020, n = 635.**

|  | F1 | F2 | F3 |
|---|---|---|---|
| **Functional end-of-life health literacy** |  |  |  |
| Understanding specialized vocabulary—prognosis | 0.72 |  |  |
| Understanding specialized vocabulary—intubation | 0.75 |  |  |
| Understanding specialized vocabulary—palliative care | 0.87 |  |  |
| Understanding specialized vocabulary—cardiopulmonary resuscitation | 0.83 |  |  |
| Understanding specialized vocabulary—artificial nutrition | 0.78 |  |  |
| Understanding specialized vocabulary—sedation | 0.66 |  |  |
| **Interactive end-of-life health literacy** |  |  |  |
| Feel comfortable—defining overtreatment |  | 0.48 |  |
| Feel comfortable—talking about end-of-life preferences |  | 0.77 |  |
| Feel comfortable—talking about end-of-life treatments |  | 0.80 |  |
| Feel comfortable—finding advance directive forms |  | 0.52 |  |
| Feel comfortable—making decisions based on probabilities |  | 0.70 |  |
| Feel comfortable—choosing treatment type if terminal disease |  | 0.71 |  |
| Feel comfortable—defining conditions when to be left to die |  | 0.71 |  |
| **Critical end-of-life health literacy** |  |  |  |
| Treatment preference regarding breathing machines |  |  | 0.82 |
| Treatment preference regarding artificial nutrition |  |  | 0.84 |
| Treatment preference regarding blood transfusions |  |  | 0.87 |
| Treatment preference regarding antibiotics |  |  | 0.82 |
| Treatment preference regarding cardiopulmonary resuscitation |  |  | 0.82 |

Rotated factor loadings for each component of the S-EOL-HLS.

scores on both literacy scales. Finally, compared to respondents who self-reported bad health, those who reported good or excellent health were more likely to have higher end-of-life health literacy and health literacy scores. The two partial associations between the scales that differed were living area and age group: respondents living in a rural area compared to an urban one and those aged 75 years and older compared to younger age group were more likely to have a lower score of end-of-life health literacy; the association was not statistically significant for the health literacy measure.

In addition, Fig 3 presents the respective average scores from the HLS-EU-Q16 and the S-EOL-HLS by categories of end-of-life care planning outcomes. Respondents who had discussed end-of-life wishes issued advance directives, or appointed a surrogate had systematically higher end-of-life health literacy and health literacy scores. However, the differences in average scores were statistically significant only for the S-EOL-HLS.

The results from the ROC analysis comparing the performance of the HLS-EU-Q16 and the S-EOL-HLS on the three end-of-life care planning outcomes are presented in Fig 4. The areas under the curve from the S-EOL-HLS were higher for all three end-of-life planning outcomes. In addition, the score from S-EOL-HLS seemed to be better distributed along the curve, which indicates a better performance.

## Discussion

Aging populations with more complex health conditions value health literacy and end-of-life health literacy as important public health issues [49, 50]. Recognizing individuals with limited

**Table 3. Partial associations of health literacy and subjective end-of-life health literacy percentage scores with respondents' sociodemographic characteristics, adults aged 50+, SHARE Switzerland, 2019/2020, n = 1,270.**

| | HLS-EU-Q16 | S-EOL-HLS |
|---|---|---|
| **Gender** (male) | | |
| female | 4.16*** | 7.47*** |
| | (0.95) | (1.29) |
| **Age group** (58–64 years) | | |
| 65–74 years | 0.12 | -1.77 |
| | (1.15) | (1.59) |
| 75+ years | -1.30 | -4.30* |
| | (1.36) | (1.89) |
| **Education** (low) | | |
| middle | 3.44* | 10.47*** |
| | (1.56) | (2.23) |
| high | 7.06*** | 16.62*** |
| | (1.67) | (2.46) |
| **Partnership status** (has a partner) | | |
| no partner | 0.36 | -0.64 |
| | (1.25) | (1.72) |
| **Make ends meet** (easily) | | |
| fairly easily | -1.01 | -1.01 |
| | (1.09) | (1.57) |
| with difficulty | -6.46*** | -7.45** |
| | (1.94) | (2.68) |
| **Language** (German (ch)) | | |
| French (ch) | 1.01 | 2.20 |
| | (1.22) | (1.72) |
| Italian (ch) | -4.12 | -7.13 |
| | (3.36) | (5.24) |
| **Living area** (urban) | | |
| rural | -1.08 | -4.07** |
| | (0.97) | (1.37) |
| **Self-rated health** (bad health) | | |
| good health | 8.58*** | 4.94* |
| | (1.74) | (2.22) |
| very good/excellent health | 11.85*** | 11.58*** |
| | (1.73) | (2.18) |
| Constant | 71.90*** | 58.91*** |
| | (2.65) | (3.41) |
| Observations | 1270 | 1270 |

Note, this table shows two Ordinary Least Squares (OLS) regressions of the European Health Literacy Scale (HLS-EU-Q16) percentage score and the Subjective End-Of-Life Health Literacy (S-EOL-HLS) percentage score on the covariates. The table shows the estimates and standard errors in brackets with significance level

* $p < 0.05$,

** $p < 0.01$,

*** $p < 0.001$.

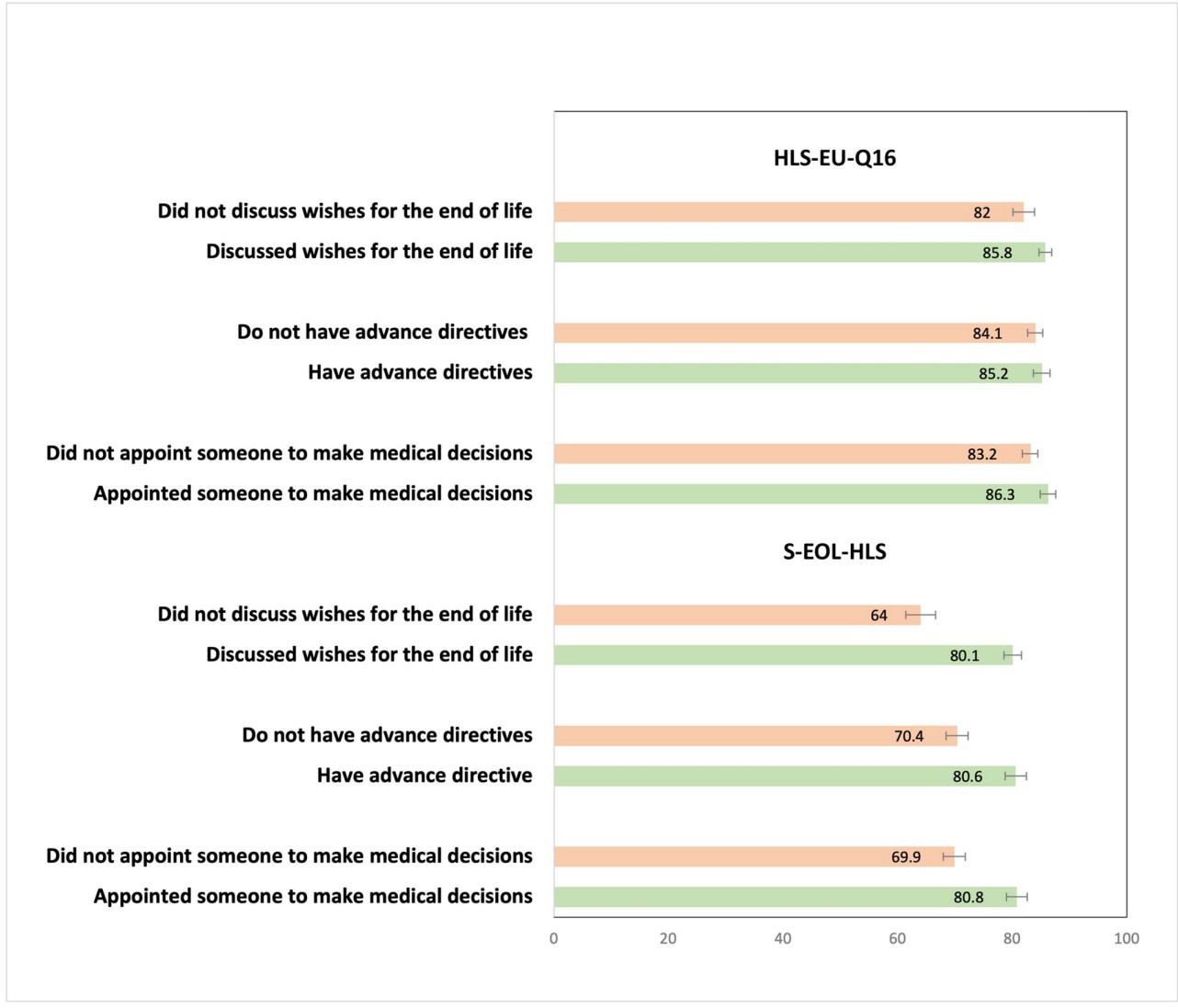

**Fig 3. Comparison of the HLS-EU-Q16 and the S-EOL-HLS percentage scores on end-of-life health outcomes, adults aged 50+, SHARE Switzerland, 2019/2020, n = 1,270.**

end-of-life health literacy and enhancing their skills to navigate specific end-of-life healthcare challenges has the potential to bolster their communication and care decision-making capacities. This study is, to the best of our understanding, pioneering in the field of end-of-life health literacy, introducing the first instrument specifically designed and validated to assess individuals' self-perceived abilities to manage end-of-life medical situations. The S-EOL-HLS allows us to draw a comprehensive picture of subjective end-of-life health literacy by measuring individuals' levels of functional, interactive, and critical end-of-life health literacy. The exploratory and confirmatory factor analysis showed that all the fit indices obtained for our samples are within acceptable limits [51]. The reliable and consistent results from the statistical validation thus showed that the S-EOL-HLS is a reliable and valid instrument to measure the end-of-life health literacy of older adults.

The S-EOL-HLS is constructed to ensure comparability with a widely established and validated general health literacy scale (HLS-EU-Q16) [35–37]. When we compared the

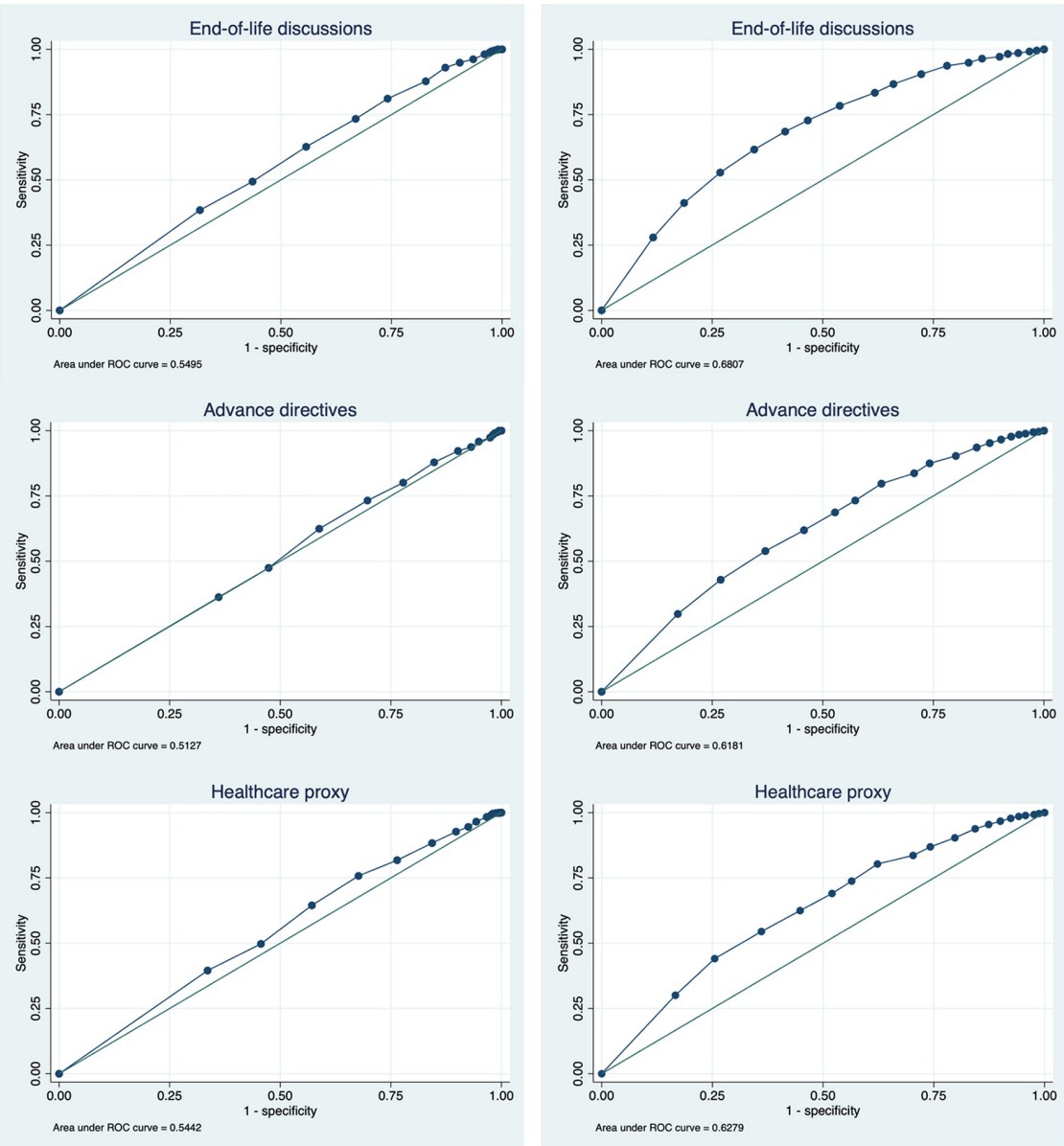

**Fig 4. Comparison of the HLS-EU-Q16 and the S-EOL-HLS on end-of-life health outcomes, ROC analysis, adults aged 50+, SHARE Switzerland, 2019/2020, n = 1270.**

discriminant ability of the two scales with regard to social, regional, and health characteristics, we found similar partial associations. These results show that our scale has a sensitivity equivalent to the HLS-EU-Q16 in identifying differences in health literacy levels between social groups. Respondents with a lower end-of-life health literacy score were significantly less likely to have engaged in end-of-life planning behavior. Conversely, individuals who had already engaged in end-of-life care planning found it easier to position themselves on end-of-life care issues. These findings show that the S-EOL-HLS performs better than the HLS-EU-Q16 in detecting individuals who have already engaged in end-of-life care planning for themselves,

demonstrating that the goal of the scale is being met. The equal performance of the S-EOL-HLS on detecting social groups with better health literacy than the HLS-EU-Q16 is an additional element to the statistical/metric validation of our scale that indicates that our scale is a reliable and stable measure of end-of-life health literacy. The S-EOL-HLS thus could help to evaluate the level of competencies of older adults to deal with end-of-life care situations.

Differences in responses to the S-EOL-HLS can emerge based on respondents' social backgrounds, geographical locations, and health conditions. For instance, applying the S-EOL-HLS among relatively healthy older adults aged 50+ living in Switzerland can illuminate their perceptions and misconceptions about end-of-life care. This information is valuable in designing proactive guidance and early interventions to enhance end-of-life literacy, facilitating greater preparedness when facing end-of-life decisions. Using the S-EOL-HLS across different population sub-groups may lead to varying results, but each result set offers equally valuable insights.

In terms of practical implications, the S-EOL-HLS could be embedded within larger surveys or administered as a standalone measure, helping to identify segments of the population with sub-optimal end-of-life health literacy. The resulting data can be used to formulate targeted educational strategies and interventions aimed at enhancing their understanding of, and engagement with, end-of-life care and decision-making. Furthermore, the S-EOL-HLS could serve as a valuable instrument in evaluating the effectiveness of interventions that target end-of-life health literacy. By gauging an individual's end-of-life health literacy before and after an intervention, we can quantitatively assess the intervention's success and guide subsequent refinements to enhance its impact. As such, the S-EOL-HLS represents a promising tool for both evaluating and improving end-of-life health literacy, acting as a potential catalyst for more informed and engaged decision-making around end-of-life care.

Further research could use this instrument to test whether individuals with lower end-of-life health literacy are more at risk of being disadvantaged in their quest for goal-concordant care at the end of life. With goal-concordant care being end-of-life care that aligns with an individual's values, preferences, and goals. Individuals with low end-of-life health literacy might have more difficulties understanding key concepts of end-of-life medicine, stating their preferences for end-of-life care, making informed medical decisions, and communicating them, which may prevent them from receiving end-of-life care and treatments in conformity with their wishes. The result could be overtreatment, undertreatment, or inappropriate treatment. Establishing someone's quest for goal-concordant care could be interpreted as individuals' active steps towards ensuring their care aligns with their goals. These steps may involve discussions with healthcare providers, completion of advance care planning documents, or conveying their wishes to family members. Therefore, in conjunction with the S-EOL-HLS assessment, complementary methods could include interviews or surveys involving patients and their family members, or reviewing medical records to identify documented discussions about care goals and corresponding treatment decisions.

The S-EOL-HLS tool could also help assess the impact of interventions aimed at improving end-of-life health literacy. This process would involve the identification of specific strategies, educational programs, or communication initiatives designed to bolster end-of-life health literacy and to measure individuals' end-of-life health literacy pre- and post-intervention studies to assess the effectiveness of such strategies. As individuals become more proficient in understanding, discussing, and making decisions about end-of-life care, they may also become more successful in articulating and pursuing their care goals. This could potentially lead to a higher prevalence of goal-concordant care and help healthcare providers and families to make decisions that better align with the patient's wishes. Evaluating the relationship between improved end-of-life health literacy and the realization of goal-concordant care could yield crucial insights into how best to support individuals during the end-of-life decision-making process.

Additionally, understanding the dynamic nature of end-of-life health literacy presents another potential research area. The S-EOL-HLS could be employed longitudinally to monitor how an individual's end-of-life health literacy evolves over time, particularly in response to significant life events such as receiving a serious illness diagnosis for oneself or a loved one. Such research could uncover trends or patterns in the evolution of end-of-life health literacy, which could in turn inform the development of interventions. Furthermore, understanding how changes in end-of-life health literacy impact the pursuit of goal-concordant care could elucidate the longitudinal relationship between these two constructs, thereby helping to identify optimal moments to support individuals in their quest for end-of-life care that aligns with their preferences.

Finally, while the S-EOL-HLS was designed to test the literacy level regarding one's own health care at the end of life, it could also be useful to develop a tool that measures the literacy to make surrogate end-of-life decisions on behalf of others, commonly family members. In fact, while everyone will only be confronted once in a life with end-of-life decisions regarding oneself, we are usually called multiple times in life to make surrogate end-of-life decisions for others.

## Limitations

Our study may have several limitations. First, as a subjective measure, the S-EOL-HLS questionnaire could include reporting bias from respondents who under- or overestimate their actual skills. Nevertheless, subjective health literacy assessments are the most suitable for population surveys for practical reasons. In addition, we know that in some areas, subjective self-assessment is reliable; for instance, self-rated health status predicts mortality very accurately [52]. Finally, in the context of end-of-life decision-making and planning, an individual's perception of their own competence in understanding, discussing, and making informed decisions about end-of-life care can significantly influence their motivation to engage in these important activities. Subjective measures of health literacy have been shown to be positively correlated with self-efficacy [33], which is associated with healthy behavior and better health status [32, 53, 54]. Thus, we assume that individuals who feel competent will be able to engage in end-of-life care planning, in our case, where they will be accompanied and guided by trained professionals who can rectify any remaining misconceptions about end-of-life care situations.

Second, the items of the S-EOL-HLS constitute only a selection of end-of-life literacy skills, which may be incomplete or biased or have limited clinical applicability. Third, the item grouping and question formats from the S-EOL-HLS may have increased the items' correlations within each factor. However, such a design was necessary for administrating questions in an understandable and consistent manner in a self-administered paper-and-pencil questionnaire to a population of older adults. Forth, selection effects and attrition of the SHARE sample might result in representativeness issues of very old adults or individuals with bad health conditions. However, these issues are common to all longitudinal population studies, and considerable efforts are undertaken to minimize these biases in the SHARE survey. Furthermore, the response rate to the Swiss paper-and-pencil questionnaire was very high, and respondents excluded from our analytical sample did not present unexpected characteristics. Finally, it was not possible to validate the scale in the three Swiss national languages separately due to notably the low number of Italian-speaking respondents. However, preliminary analysis showed high internal consistency of the German and French versions of the questionnaire, with Cronbach's alpha of 0.92 for the German subsample and 0.94 for the French.

## Conclusion

End-of-life health literacy has become an important public health issue with the aging populations and their ensuing transformation of the last phase of life. Limited end-of-life health literacy presents an additional and substantial barrier to communication and decision-making at the end of life. In addition, Individuals more often have to make complex end-of-life medical decisions in situations of physical and mental impairment—for themselves and others. Improving individuals' abilities and proficiency to deal with situations specific to end-of-life care and medicine would empower them to initiate reflection, communication, and engagement in end-of-life care planning and decisions. This study demonstrated that the S-EOL-HLS is a reliable and valid instrument to measure older adults' self-perceived end-of-life health literacy. The S-EOL-HLS evaluates the level of comfort and competence of the general population in handling end-of-life care situations, its associations with end-of-life care planning outcomes, and its similarities with the results of the HLS-EU-Q16 support this. Future research with the S-EOL-HLS may reveal important insights. It could explore if lower S-EOL-HLS scores correlate with less alignment between patients' preferences and their received end-of-life care. The tool could also assess the impact of interventions aimed at enhancing end-of-life health literacy, offering key evaluation metrics for these initiatives. Additionally, employing the S-EOL-HLS in longitudinal studies may help elucidate how end-of-life health literacy evolves in response to major life events, such as personal or family illness diagnoses.

## Supporting information

**S1 Fig. The 3-factors model of the S-EOL-HLS.**
(TIF)

**S2 Fig. S-EOL-HLS scree plot of eigenvalues.**
(TIF)

**S1 Table. The 18 items from the S-EOL-HLS scale.**
(DOCX)

**S2 Table. The 16 items from the HLS-EU-Q16 scale.**
(DOCX)

## Acknowledgments

We would like to extend our sincere gratitude to Dr. Valérie-Anne Ryser for her invaluable advice throughout the preparation of this paper. Her profound insights, academic rigor, and unwavering support have played a critical role in shaping this research. Moreover, her assistance in the preparation of our presentation at the GSA 2023 conference greatly contributed to its success. Her collaboration has not only significantly enriched this work but also our personal growth as researchers. We look forward to the opportunity of collaborating with her in the future.

## Author Contributions

**Conceptualization:** Clément Meier, Sarah Vilpert, Carmen Borrat-Besson, Ralf J. Jox, Jürgen Maurer.

**Formal analysis:** Clément Meier, Maud Wieczorek, Jürgen Maurer.

**Funding acquisition:** Sarah Vilpert, Jürgen Maurer.

**Investigation:** Jürgen Maurer.

**Methodology:** Clément Meier, Maud Wieczorek.

**Writing – original draft:** Clément Meier, Jürgen Maurer.

**Writing – review & editing:** Sarah Vilpert, Maud Wieczorek, Carmen Borrat-Besson, Ralf J. Jox, Jürgen Maurer.

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
