## [Decision Letter · Decision Letter 0]

16 Jun 2023

PONE-D-23-11080Development and validation of a subjective end-of-life health literacy scalePLOS ONE

Dear Dr. Meier,

Thank you for submitting your manuscript to PLOS ONE. After careful consideration, we feel that it has merit but does not fully meet PLOS ONE’s publication criteria as it currently stands. Therefore, we invite you to submit a revised version of the manuscript that addresses the points raised during the review process. Please submit your revised manuscript by Jul 31 2023 11:59PM. If you will need more time than this to complete your revisions, please reply to this message or contact the journal office at plosone@plos.org. Please include the following items when submitting your revised manuscript:A rebuttal letter that responds to each point raised by the academic editor and reviewer(s). You should upload this letter as a separate file labeled 'Response to Reviewers'.A marked-up copy of your manuscript that highlights changes made to the original version. You should upload this as a separate file labeled 'Revised Manuscript with Track Changes'.An unmarked version of your revised paper without tracked changes. You should upload this as a separate file labeled 'Manuscript'.

We look forward to receiving your revised manuscript.

Kind regards,

Stefano Occhipinti

Academic Editor

PLOS ONE

Journal Requirements:

"SNSF funding for the end-of-life project. Healthy Ageing in the Face of Death: Preferences, Communication, Knowledge, and Behaviors Regarding End of Life and End-of-life Planning Among Older Adults in Switzerland (grant number: 10001C_188836)."

"The SHARE data collection has been funded by the European Commission, DG RTD through FP5 (QLK6-CT-2001-00360), FP6 (SHARE-I3: RII-CT-2006- 062193, COMPARE: CIT5-CT-2005-028857, SHARELIFE: CIT4-CT-2006-028812), FP7 (SHAREPREP: GA N°211909, SHARE-LEAP: GA N°227822, SHAREM4: GA N°261982, DASISH: GAN°283646) and Horizon 2020 (SHARE-DEV3: GA N°676536, SHARE-COHESION: GAN°870628, SERISS: GA N°654221, SSHOC: GA N°823782, SHARE-COVID19: GA N°101015924) and by DG Employment, Social Affairs & Inclusion through VS 2015/0195, VS 2016/0135, VS 2018/0285, VS 2019/0332, and VS 2020/0313. Additio nal funding from the German Ministry of Education and Research, the Max Planck Society for the Advancement of Science, the U.S. National Institute on Aging (U01_AG09740-13S2, P01_AG005842, P01_AG08291, P30_AG12815, R21_AG025169, Y1-AG-4553-01, IAG_BSR06-11, OGHA_04-064, HHSN271201300071C, RAG052527A) and from various national funding sources is gratefully acknowledged (see https://share-eric.eu)."

"SNSF funding for the end-of-life project. Healthy Ageing in the Face of Death: Preferences, Communication, Knowledge, and Behaviors Regarding End of Life and End-of-life Planning Among Older Adults in Switzerland (grant number: 10001C_188836)."

**Additional Editor Comments:**

The paper was read by two reviewers and myself. All of us are generally positive about the paper. However, Reviewer 2 has raised a number of important points about the communication, and especially, the contextualisation of the results. I concur and believe the paper would be much stronger if you could address these points. You will see that the issues raised are primarily about the way the work is presented. I would urge you to address each of these points in turn and submit the revision for further consideration.

Reviewers' comments:

Reviewer's Responses to Questions

**Comments to the Author**

1. Is the manuscript technically sound, and do the data support the conclusions?

Reviewer #1: Yes

Reviewer #2: Partly

2. Has the statistical analysis been performed appropriately and rigorously? 

Reviewer #1: Yes

Reviewer #2: I Don't Know

3. Have the authors made all data underlying the findings in their manuscript fully available?

Reviewer #1: Yes

Reviewer #2: Yes

4. Is the manuscript presented in an intelligible fashion and written in standard English?

Reviewer #1: Yes

Reviewer #2: Yes

5. Review Comments to the Author

Reviewer #1: Congratulation for the work you have done. The devised scale is a necessary and important tool for data collection about a sensitive health related issue. All details regarding the methodology had been used were provided and interpretation of the findings are sound and logical. No further revision is required.

Reviewer #2: This article presents an interesting study detailing the development of a survey instrument for use in the measurement of three components of health literacy related to end-of-life care and decision making. The topic is relevant and the findings of such a study and the associated tool development could be very useful. The manuscript is usually clearly written.

Overall, however, I did not get a clear picture of how the survey tool itself may contribute to improving population level EOL health literacy. I do not think that the following questions were addressed by the authors: How and when would this tool be used? How would the findings of such a survey translate to improvements in end-of-life care or intervention planning?

The article would benefit from some more context about the location of the study – for example, brief comments on cultural or social factors that may influence the survey results – e.g. what legal status do advance directives have? How much public education has been undertaken already in region? These factors vary widely across different geographical locations and are relevant to these types of studies.

I also found that the article discussed very broad target groups – that is, people in clinical settings as well as the general population. These groups are very different and are likely to respond to various questions differently. Also, there is evidence that people respond differently to end-of-life issues when they (or family members) are actually in a crisis situation. Therefore, how useful is data collected from healthy individuals in relation to end-of-life service planning? The authors discuss the usefulness of the tool in relation to advance care planning and advance directives – this could be clarified further by giving some additional context (e.g. legal status of advance directives) and discussion around how the results of such a survey could be used (i.e. to help design public education programs etc).

Can the authors make some specific suggestions or clearer links with future research? The following suggestion for future research are made in the conclusion:

“Further research using the S-EOL-HLS is needed to test the hypothesis that individuals with a low end-of-life health literacy are at risk of being disadvantaged in their quest for goal-concordant care at the end of life.”

Specifically what sort of research are the authors suggesting could be done and what would be outcomes/contribution? The wording “goal-concordant care at the end-of-life” is ambiguous in this context – how would you measure this and how would you establish someone’s “quest” for such care?

The following comments relate to specific sections of the article (page numbers correspond with those shown in the manuscript):

The writing style is generally clear but could be further improved. Some sentences are very long and could be reworded/rewritten as shorter sentences for clarity. These are some examples:

(Introduction) “Health literacy skills are particularly important in aging populations with a higher prevalence of chronic diseases and the frequent need for potentially complex treatment regimes in cases of multimorbidity, as health literacy influences how people with chronic diseases perceive their health challenges, communicate with healthcare providers, and make medical decisions”

(Page 6) “Importantly for our context, subjective measures of health literacy also appear to be closely related to the concept of self-efficacy applied to the completion of relevant health literacy tasks (28–30), which makes measurement of subjective health literacy interesting in its own right, as self-efficacy in health literacy tasks, i.e., individuals’ judgment of their capabilities to execute specific health literacy tasks may by itself be an important determinant of health behaviors, healthcare use, and related outcomes”

There are inconsistencies in the use of hyphens in the term end of life versus end-of-life.

Page 4: I am not sure how the discussion of death literacy contributed to the surrounding discussion

Page 5: Make a new paragraph with the research aims for clarity.

Page 5: sentence under conceptual framework needs to be corrected:

“(2) to effectively engaging, interacting and applying new information with healthcare providers and family concerning advance care planning and end-of-life care (interactive health literacy)”

Page 7: The survey sets out to cover a large number and range of topics. The items related to applying the new information and critical end-of-literacy cover extremely complex facets of knowledge and decision-making. Responses to these items may vary significantly depending on when an individual completes the survey (i.e. when they are healthy and at home compared to when someone is in hospital and faced with these choices in reality).

In terms of “applying new information” – this wording suggests that the respondent may be given new information about these topics prior to completing the next part of the survey. What “new information” are the authors referring to?

The results section says that 84% of respondents reported good or excellent health – how does this health status effect their responses to these types of questions? If the data generated by the survey aims to improve public health campaigns then this is also of note and can be discussed further.

Page 10: “Respondents felt less comfortable when they had to make decisions based on probabilities (39,7%), define overtreatment (34.6%), define specific conditions or situations in which to die (37.7%), and choose the type of treatment in case of a terminal disease (36.1%).”

Discussion

The discussion needs to be refined further as it currently covers a very broad range of issues/topics and the links between these are not clear. For example, how does a population level survey of generally healthy people link with actual choices/preferences/behaviours in a health crisis?

The following sentence is very broad and could be refined further:

“Identifying patients with limited end-of-life health literacy and improving their abilities to deal with the specific healthcare challenges at the end of life could empower them in their communication and care decision-making. To the best of our knowledge, our population-based study of end-of life health literacy is the first to develop and validate a specific instrument for assessing individuals’ self-perceived competencies to deal with end-of-life medical situations.” (page 12)

The following point could also be examined in more detail and it is also unsurprising - “Respondents with a lower end-of-life health literacy score were significantly less likely to have engaged in end-of-life planning behavior”.

Limitations

This sentence is far too broad – can the authors demonstrate how this applies to the end-of-life context? What is meant by “competence”?

“Finally, the perception of one’s own competence is precisely what will motivate or demotivate an individual to undertake an action, whatever its scope.”

6. PLOS authors have the option to publish the peer review history of their article (what does this mean?). If published, this will include your full peer review and any attached files.

Reviewer #1: **Yes: **Professor Abdolreza Shaghaghi

Reviewer #2: No

---

## [Author Response · Author response to Decision Letter 0]

24 Jul 2023

Dear Edrian Nim Tolentino,

I appreciate the time you have taken to review our submission and provide detailed feedback. 

We appreciate your effort to align the Financial Disclosure Statement with PLOS conventions. However, I would like to point out a minor inaccuracy regarding the grant recipient. The grant (10001C_188836) was awarded to Professor Jürgen Maurer, under whose guidance I have been employed for this end-of-life project. Consequently, the revised Financial Disclosure Statement should read:

"This study was funded by the Swiss National Science Foundation (SNSF) for the end-of-life project. Healthy Ageing in the Face of Death: Preferences, Communication, Knowledge, and Behaviors Regarding End of Life and End-of-life Planning Among Older Adults in Switzerland (grant number: 10001C_188836) awarded to Prof. JM, CM is employed for this project. The funders had no role in study design, data collection, and analysis, decision to publish, or preparation of the manuscript."

I trust that this amendment accurately reflects the project's funding structure and appreciate your understanding on this matter. 

Thank you again for your support and guidance throughout this process. 

Best regards,

Clément Meier

Dear Prof. Occhipinti,

We are confident that the revisions have substantially enhanced the quality and rigor of our research, resulting in a more robust and comprehensive manuscript.

We would like to take this opportunity to express our sincere appreciation for the valuable input provided by the reviewers and yourself. Their insightful comments and suggestions have undoubtedly contributed to the improvement of our manuscript. We are grateful for the opportunity to address their concerns and believe that the revised manuscript now meets the high standards of PLOS ONE.

Thank you once again for your time, consideration, and the opportunity to submit our revised manuscript. We look forward to your favorable decision and hope that our research will make a valuable contribution to the scientific community.

Yours sincerely,

Clément Meier

University of Lausanne

Geopolis – FORS, 1015 Lausanne

clement.meier@unil.ch

Clément Meier

University of Lausanne

Geopolis – FORS, 1015 Lausanne

clement.meier@unil.ch

Prof. Stefano Occhipinti

Academic Editor

PLOS ONE

Lausanne, the 10th of July 2023

Dear Prof. Occhipinti,

Re: Rebuttal Letter - Manuscript ID: [PONE-D-23-11080]

We would like to express our gratitude to you and the reviewers for taking the time to review our manuscript titled "Development and Validation of a Subjective End-of-Life Health Literacy Scale." We greatly appreciate the thoughtful feedback and valuable suggestions provided, which have undoubtedly strengthened our research. In response to the comments raised, we have carefully revised the manuscript and addressed each point in detail. Please find our point-by-point rebuttal below.

Reviewer 1:

1. Congratulation for the work you have done. The devised scale is a necessary and important tool for data collection about a sensitive health related issue. All details regarding the methodology had been used were provided and interpretation of the findings are sound and logical. No further revision is required.

Thank you very much for your constructive and positive feedback on our study. We appreciate your recognition of the necessity and importance of the Subjective End-Of-Life Health Literacy Scale (S-EOL-HLS). We are also grateful for your acknowledgment of the methodology used and the interpretation of our findings. Your endorsement of our work as logical and sound is indeed very encouraging. Your comments provide significant motivation for us to continue our research in this field, and we hope our work will contribute positively to the understanding and improvement of health-related issues, particularly those surrounding the end of life. Thank you again for your time and valuable feedback.

Reviewer 2:

1. This article presents an interesting study detailing the development of a survey instrument for use in the measurement of three components of health literacy related to end-of-life care and decision making. The topic is relevant and the findings of such a study and the associated tool development could be very useful. The manuscript is usually clearly written.

We appreciate your insightful feedback on our article and for recognizing the relevance and potential usefulness of the S-EOL-HLS. We are pleased to read that you find the study interesting and the tool to be of potential utility in the measurement of health literacy related to end-of-life care and decision making. We also value your comment about the clarity of our manuscript's presentation. Thank you for your time and the positive appraisal of our work.

2. Overall, however, I did not get a clear picture of how the survey tool itself may contribute to improving population level EOL health literacy. I do not think that the following questions were addressed by the authors: How and when would this tool be used? How would the findings of such a survey translate to improvements in end-of-life care or intervention planning?

Thank you for your thoughtful comments and questions. You have raised an important point about the practical implications of the S-EOL-HLS for improving end-of-life health literacy at the population level. This tool was developed primarily as a means to quantify and assess an individual's capacity to understand and engage with end-of-life health information and decision making. While the current study was focused on its development and validation, its practical application extends far beyond this. The S-EOL-HLS can be incorporated into broader surveys or used as a standalone tool to identify population groups who may lack adequate end-of-life health literacy. This information can guide targeted education and intervention strategies to enhance their understanding and engagement with end-of-life care and decision making. The tool may also be employed in pre- and post-intervention studies to assess the effectiveness of such strategies in improving end-of-life health literacy. Also, the scale can be used whenever there is a need to assess end-of-life health literacy, whether for research, intervention planning, or evaluation purposes. Given the sensitivity of the subject matter, the timing of its use should be considered carefully in relation to an individual's health status and readiness to engage with end-of-life issues. We hope this response adequately addresses your questions; we added the following clarifications in discussion section from the manuscript to better articulate these points. 

"In terms of practical implications, the S-EOL-HLS could be embedded within larger surveys or administered as a standalone measure, helping to identify segments of the population with sub-optimal end-of-life health literacy. The resulting data can be used to formulate targeted educational strategies and interventions aimed at enhancing their understanding of, and engagement with, end-of-life care and decision-making. Furthermore, the S-EOL-HLS could serve as a valuable instrument in evaluating the effectiveness of interventions that target end-of-life health literacy. By gauging an individual's end-of-life health literacy before and after an intervention, we can quantitatively assess the intervention's success and guide subsequent refinements to enhance its impact. As such, the S-EOL-HLS represents a promising tool for both evaluating and improving end-of-life health literacy, acting as a potential catalyst for more informed and engaged decision-making around end-of-life care."

3. The article would benefit from some more context about the location of the study – for example, brief comments on cultural or social factors that may influence the survey results – e.g. what legal status do advance directives have? How much public education has been undertaken already in region? These factors vary widely across different geographical locations and are relevant to these types of studies.

We thank you for your insightful suggestion to provide more context about the location of our study, acknowledging that cultural, social, and legal factors can influence survey results. We agree that these aspects are integral to our study, and we incorporated them into the revised manuscript. Regarding the legal status of advance directives in Switzerland, they are legally binding, provided they meet the requirements stipulated in the Swiss Civil Code. Previous studies by our research group showed that the actual public awareness and completion of advance directives remain low in the Swiss population, despite various efforts by the authorities to increase public information on the subject. We have now provided more detailed information on these aspects within the manuscript. It's important to note that while our study is based in Switzerland, we envisage the S-EOL-HLS could be adapted for use in other geographical and cultural contexts, with the necessary adjustments and validation. 

Here is the paragraph we added to the manuscript: "Before delving into the specifics of our study, it's pertinent to shed light on some region-specific factors within Switzerland, the country where our research was conducted. The legal status of Advance Directives in Switzerland has been firmly established since 2013, when the Swiss Federal Council acknowledged their importance and introduced a new adult protection law into the Swiss Civil Code [15]. This significant legislative step amplified public awareness around end-of-life planning, however, despite this new law and the general growth of interest regarding end-of-life issues and palliative care, research indicates that the proportion of the general population completing advance directives remains relatively low [16]. Furthermore, it's crucial to acknowledge Switzerland's significant linguistic diversity, with the country partitioned into three main regions: German, French, and Italian. Differences across these regions are often observed, notably in the varying preferences, attitudes, and behaviors related to end-of-life care and planning [16,17]."

Thank you for highlighting the point regarding regional differences. In our study, we have indeed taken these potential regional disparities into account, as we investigated if the end-of-life health literacy scores varied between different linguistic regions, the results of which are presented in Table 3. 

4. I also found that the article discussed very broad target groups – that is, people in clinical settings as well as the general population. These groups are very different and are likely to respond to various questions differently. Also, there is evidence that people respond differently to end-of-life issues when they (or family members) are actually in a crisis situation. Therefore, how useful is data collected from healthy individuals in relation to end-of-life service planning? 

Thank you for pointing out the potential differences in response to end-of-life issues among different target groups and under varying circumstances. You have correctly highlighted a complex and critical aspect of end-of-life research. The S-EOL-HLS is designed to measure individuals' perceived competencies to deal with end-of-life medical situations, irrespective of their health status. While responses may indeed vary between clinical populations and the general public, or during crisis situations, each perspective provides valuable insights. Data collected from healthy individuals can inform us about their understanding, anticipations, and potential misconceptions about end-of-life care and decision-making before they encounter a crisis situation. This knowledge can be used to guide the design of anticipatory guidance and early interventions to enhance end-of-life literacy in the general population, ultimately supporting individuals to be better prepared when they or their family members face end-of-life issues. However, we do acknowledge that applying the scale in a clinical setting or during a crisis might yield different, yet equally informative, data. These data could be valuable for evaluating and refining clinical practices and emergency interventions. 

We added further explanations following your important comment: "Differences in responses to the S-EOL-HLS can emerge based on respondents' social backgrounds, geographical locations, and health conditions. For instance, applying the S-EOL-HLS among relatively healthy older adults aged 50+ living in Switzerland can illuminate their perceptions and misconceptions about end-of-life care. This information is valuable in designing proactive guidance and early interventions to enhance end-of-life literacy, facilitating greater preparedness when facing end-of-life decisions. Using the S-EOL-HLS across different population sub-groups may lead to varying results, but each result set offers equally valuable insights". 

5. The authors discuss the usefulness of the tool in relation to advance care planning and advance directives – this could be clarified further by giving some additional context (e.g. legal status of advance directives) and discussion around how the results of such a survey could be used (i.e. to help design public education programs etc).

Thank you for your insightful feedback on the potential role of the S-EOL-HLS in relation to advance care planning and advance directives. We agree that further clarification and context would enhance our discussion on this topic. As mentioned in points 2 and 3, we now provide new information in the revised manuscript regarding the Swiss context and additional discussion on how the survey results could guide public education program design.

6. Can the authors make some specific suggestions or clearer links with future research? The following suggestion for future research are made in the conclusion: "Further research using the S-EOL-HLS is needed to test the hypothesis that individuals with a low end-of-life health literacy are at risk of being disadvantaged in their quest for goal-concordant care at the end of life."

Thank you for your comment on the need for clearer links with future research. We agree that more specific suggestions would enrich the discussion of our study. We envisage a few key directions for future research using the S-EOL-HLS. One would be to investigate the correlation between S-EOL-HLS scores and individuals' experiences of end-of-life care. Are individuals with lower S-EOL-HLS scores less likely to receive care that aligns with their wishes? This would indeed test the hypothesis mentioned in our conclusion. Another research direction could be to evaluate the impact of interventions aimed at improving end-of-life health literacy. Such studies could use the S-EOL-HLS to measure changes in end-of-life health literacy pre- and post-intervention. Furthermore, longitudinal studies using the S-EOL-HLS could explore how end-of-life health literacy evolves over time and in response to life events, such as the diagnosis of a serious illness in oneself or a family member. We included these specific suggestions in our revised manuscript and hope they will inspire fruitful research endeavors.

We revised the sentence you mentioned and added this one to the manuscript: "Future research with the S-EOL-HLS may reveal important insights. It could explore if lower S-EOL-HLS scores correlate with less alignment between patients' preferences and their received end-of-life care. The tool could also assess the impact of interventions aimed at enhancing end-of-life health literacy, offering key evaluation metrics for these initiatives. Additionally, employing the S-EOL-HLS in longitudinal studies may help elucidate how end-of-life health literacy evolves in response to major life events, such as personal or family illness diagnoses."

7. Specifically what sort of research are the authors suggesting could be done and what would be outcomes/contribution? The wording "goal-concordant care at the end-of-life" is ambiguous in this context – how would you measure this and how would you establish someone's "quest" for such care?

Thank you for your constructive comments. We agree that we need to be more specific about potential research avenues and intended outcomes. When we refer to "goal-concordant care at the end-of-life," we mean care that aligns with the individual's values, preferences, and goals. It is a widely used term in palliative and end-of-life care research and practice, but we understand that it can be ambiguous without appropriate context. Measuring goal-concordant care can indeed be challenging, as it requires understanding individuals' personal goals for their care and then evaluating to what extent the care they receive aligns with these goals. Methods for assessing this may include interviews or surveys with patients or their family members, or review of medical records for documented discussions about care goals and subsequent treatment decisions. As for the "quest" for such care, this could be interpreted as individuals' active steps to ensure their care aligns with their goals, such as discussions with healthcare providers, completion of advance care planning documents, or communication with family members about their wishes. 

We incorporated your comments and added more information in the discussion section of the manuscript: 

"Further research could use this instrument to test whether individuals with lower end-of-life health literacy are more at risk of being disadvantaged in their quest for goal-concordant care at the end of life. With goal-concordant care being end-of-life care that aligns with an individual's values, preferences, and goals. Individuals with low end-of-life health literacy might have more difficulties understanding key concepts of end-of-life medicine, stating their preferences for end-of-life care, making informed medical decisions, and communicating them, which may prevent them from receiving end-of-life care and treatments in conformity with their wishes. The result could be overtreatment, undertreatment, or inappropriate treatment. Establishing someone's quest for goal-concordant care could be interpreted as individuals' active steps towards ensuring their care aligns with their goals. These steps may involve discussions with healthcare providers, completion of advance care planning documents, or conveying their wishes to family members. Therefore, in conjunction with the S-EOL-HLS assessment, complementary methods could include interviews or surveys involving patients and their family members, or reviewing medical records to identify documented discussions about care goals and corresponding treatment decisions. 

The S-EOL-HLS tool could also help assess the impact of interventions aimed at improving end-of-life health literacy. This process would involve the identification of specific strategies, educational programs, or communication initiatives designed to bolster end-of-life health literacy and to measure individuals' end-of-life health literacy pre- and post-intervention studies to assess the effectiveness of such strategies. As individuals become more proficient in understanding, discussing, and making decisions about end-of-life care, they may also become more successful in articulating and pursuing their care goals. This could potentially lead to a higher prevalence of goal-concordant care and help healthcare providers and families to make decisions that better align with the patient's wishes. Evaluating the relationship between improved end-of-life health literacy and the realization of goal-concordant care could yield crucial insights into how best to support individuals during the end-of-life decision-making process.

Additionally, understanding the dynamic nature of end-of-life health literacy presents another potential research area. The S-EOL-HLS could be employed longitudinally to monitor how an individual's end-of-life health literacy evolves over time, particularly in response to significant life events such as receiving a serious illness diagnosis for oneself or a loved one. Such research could uncover trends or patterns in the evolution of end-of-life health literacy, which could in turn inform the development of interventions. Furthermore, understanding how changes in end-of-life health literacy impact the pursuit of goal-concordant care could elucidate the longitudinal relationship between these two constructs, thereby helping to identify optimal moments to support individuals in their quest for end-of-life care that aligns with their preferences."

8. The following comments relate to specific sections of the article (page numbers correspond with those shown in the manuscript): The writing style is generally clear but could be further improved. Some sentences are very long and could be reworded/rewritten as shorter sentences for clarity. These are some examples: (Introduction) "Health literacy skills are particularly important in aging populations with a higher prevalence of chronic diseases and the frequent need for potentially complex treatment regimes in cases of multimorbidity, as health literacy influences how people with chronic diseases perceive their health challenges, communicate with healthcare providers, and make medical decisions". (Page 6) "Importantly for our context, subjective measures of health literacy also appear to be closely related to the concept of self-efficacy applied to the completion of relevant health literacy tasks (28–30), which makes measurement of subjective health literacy interesting in its own right, as self-efficacy in health literacy tasks, i.e., individuals' judgment of their capabilities to execute specific health literacy tasks may by itself be an important determinant of health behaviors, healthcare use, and related outcomes"

Thank you for your valuable feedback. We appreciate your suggestion to revise some of the longer sentences for clarity and understand that shorter sentences can improve readability. We have edited the sentences you highlighted. Here are the revised versions:

"Health literacy skills hold particular importance in aging populations where chronic diseases are more prevalent. These skills are essential due to the frequent need for complex treatment regimes, especially in cases of multimorbidity. Health literacy significantly influences how individuals perceive their health challenges, communicate with healthcare providers, and make medical decisions."

"Subjective measures of health literacy have a notable connection with the concept of self-efficacy in health literacy tasks. This connection makes the measurement of subjective health literacy interesting. Specifically, individuals' judgments of their capabilities to execute specific health literacy tasks—i.e., self-efficacy—may be a significant determinant of health behaviors, healthcare use, and related outcomes."

9. There are inconsistencies in the use of hyphens in the term end of life versus end-of-life.

Thank you for pointing out the inconsistent use of hyphens with the term "end of life." We appreciate your attention to detail. We went through the manuscript thoroughly and corrected these inconsistencies and ensure uniform usage.

10. Page 4: I am not sure how the discussion of death literacy contributed to the surrounding discussion

Thank you for your valuable feedback. We mention "death literacy" to highlight an existing concept in the field of end-of-life care research that is somewhat related to our study, but not exactly the same as our concept of "end-of-life health literacy". "Death literacy" refers to knowledge and skills related to understanding and navigating the death system, including elements such as factual knowledge, emotional support, and community capacity. We aimed to illustrate that while "death literacy" is an important concept, it does not encapsulate the more specific health-related competencies and decision-making abilities at the end-of-life that we aim to measure with the S-EOL-HLS. However, we acknowledge that this connection may not have been clear in our original manuscript. In our revision, we clarified the relationship between "death literacy" and our concept of "end-of-life health literacy", emphasizing how our scale complements existing tools and frameworks by focusing specifically on the unique competencies required to navigate medical decisions at the end-of-life. Thank you once again for your insightful comment. We added this clarification to the manuscript: 

"While the concept of death literacy encompasses knowledge and skills related to understanding and navigating the death system, it does not specifically address individual's ability to navigate medical decisions at the end of life."

11. Page 5: Make a new paragraph with the research aims for clarity.

We thank you for your suggestion, we created a new paragraph with the research aims. 

12. Page 5: sentence under conceptual framework needs to be corrected: "(2) to effectively engaging, interacting and applying new information with healthcare providers and family concerning advance care planning and end-of-life care (interactive health literacy)"

Thank you for your constructive feedback. We agree that the sentence you pointed out in the Conceptual Framework section needs to be revised for better clarity and grammatical correctness. We appreciate your attention to detail. Here is the corrected version of the sentence: 

"(2) to effectively engage, interact and apply newly-acquired information in discussions with healthcare providers and family concerning advance care planning and end-of-life care (interactive health literacy)"

13. Page 7: The survey sets out to cover a large number and range of topics. The items related to applying the new information and critical end-of-literacy cover extremely complex facets of knowledge and decision-making. Responses to these items may vary significantly depending on when an individual completes the survey (i.e. when they are healthy and at home compared to when someone is in hospital and faced with these choices in reality). 

We appreciate your thoughtful insights about the scope and complexity of topics covered by our survey. You're correct in stating that the items concerning "applying new information" and "critical end-of-life literacy" do cover intricate facets of knowledge and decision-making. We fully acknowledge that an individual's responses may indeed change based on their health status at the time of survey completion or the context in which these decisions are being considered. This is a limitation that is inherent to any attempt to measure perceived health literacy in dynamic and context-dependent topics such as end-of-life decision-making. 

14. In terms of "applying new information" – this wording suggests that the respondent may be given new information about these topics prior to completing the next part of the survey. What "new information" are the authors referring to?

Regarding your query about "applying new information," we apologize for any confusion. In our context, "new information" refers to any newly-acquired knowledge relevant to end-of-life care and advance care planning, which an individual may encounter in the course of their healthcare journey or personal research. This could be information gained from healthcare providers, self-guided research, or conversations with family and friends. The survey does not provide any new information immediately prior to the completion of these items. Our intention here is to assess how individuals perceive their ability to absorb, understand, and then apply this new knowledge when considering or making end-of-life decisions. We do not presume to offer new information during the survey, but rather evaluate how confident respondents feel in their ability to process and utilize such information when encountered. We understand how the current wording might lead to misunderstanding, and we change the sentence to make it clearer: 

"(2) to effectively engage, interact and apply newly-acquired information in discussions with healthcare providers and family concerning advance care planning and end-of-life care (interactive health literacy)"

15. The results section says that 84% of respondents reported good or excellent health – how does this health status effect their responses to these types of questions? 

Thank you for your insightful comment. You have highlighted an important aspect which is the influence of current health status on the responses to questions about end-of-life care. Indeed, we acknowledge that respondents' current health status could impact their perceptions of and responses to the end-of-life health literacy questions. As presented in Table 3, compared to respondents who self-reported bad health, those who reported good or excellent health were more likely to have higher end-of-life health literacy and health literacy scores. Specifically, individuals who report good or excellent health might not have confronted the reality of severe illness or the end of life, which could affect their perceived ease or difficulty in understanding and making decisions about end-of-life care. We hypothesize that those with poorer health or who have had more interactions with the healthcare system might have different responses compared to their healthier counterparts due to more real-world experience with these situations. We appreciate your comment and will ensure that we explore this question in our future analysis. 

16. If the data generated by the survey aims to improve public health campaigns then this is also of note and can be discussed further.

Thank you for highlighting the relevance of our study in the context of public health campaigns. You are correct in asserting that our findings can contribute significantly to the design and effectiveness of public health initiatives. We fully agree with your suggestion. As mentioned in point 7, we elaborated more on this in the discussion section. 

17. Page 10: "Respondents felt less comfortable when they had to make decisions based on probabilities (39,7%), define overtreatment (34.6%), define specific conditions or situations in which to die (37.7%), and choose the type of treatment in case of a terminal disease (36.1%)."

Thank you for your constructive feedback. We appreciate your attention to detail. Here is an updated version of the sentence: 

"Respondents reported unease in various situations: 39.7% found it difficult to make decisions that involved probabilities; 34.6% struggled to define the term 'overtreatment'; 37.7% had difficulties in specifying conditions or circumstances under which they would prefer to die; and, 36.1% were uncomfortable when asked to choose a type of treatment if they were to have a terminal illness."

18. Discussion. The discussion needs to be refined further as it currently covers a very broad range of issues/topics and the links between these are not clear. For example, how does a population level survey of generally healthy people link with actual choices/preferences/behaviours in a health crisis?

Thank you for your suggestion on refining the discussion section of our study. We acknowledge your concern that the discussion currently covers a broad range of topics and agree that we need to make the links between these topics more explicit. As addressed in our responses to Points 2, 4, and 7, additional information has been thoughtfully incorporated into the discussion section of the manuscript.

19. The following sentence is very broad and could be refined further: "Identifying patients with limited end-of-life health literacy and improving their abilities to deal with the specific healthcare challenges at the end of life could empower them in their communication and care decision-making. To the best of our knowledge, our population-based study of end-of life health literacy is the first to develop and validate a specific instrument for assessing individuals' self-perceived competencies to deal with end-of-life medical situations." (page 12)

Thank you for pointing out the broad nature of the sentence you mentioned. We appreciated your feedback and revised the sentence: 

"Recognizing individuals with limited end-of-life health literacy and enhancing their skills to navigate specific end-of-life healthcare challenges has the potential to bolster their communication and care decision-making capacities. This study is, to the best of our understanding, pioneering in the field of end-of-life health literacy, introducing the first instrument specifically designed and validated to assess individuals' self-perceived abilities to manage end-of-life medical situations."

Further information has also been incorporated into the discussion (Points 2, 4, and 7). 

20. The following point could also be examined in more detail and it is also unsurprising - "Respondents with a lower end-of-life health literacy score were significantly less likely to have engaged in end-of-life planning behavior".

Your observation regarding the significant relationship between end-of-life health literacy score and engagement in end-of-life planning is insightful. Indeed, exploring this relationship in further detail could offer meaningful contributions to the understanding of end-of-life planning behavior. However, the primary objective of this particular manuscript was to focus on the "Development and Validation of a Subjective End-of-Life Health Literacy Scale". Therefore, while acknowledging the potential richness of the relationship you highlighted, we decided to limit our focus in order to adequately detail the process and outcomes of our scale development and validation. That said, we agree with your suggestion and intend to explore this relationship more extensively in a future dedicated article. Thank you for your constructive feedback.

21. Limitations. This sentence is far too broad – can the authors demonstrate how this applies to the end-of-life context? What is meant by "competence"? "Finally, the perception of one's own competence is precisely what will motivate or demotivate an individual to undertake an action, whatever its scope."

Thank you for your insightful comment. Indeed, our use of the term "competence" was too broad and lacked context. In the context of our study, "competence" refers to an individual's self-perception of their ability or readiness to undertake actions related to end-of-life decision-making, such as understanding medical terms, engaging in discussions about treatment preferences, or preparing advance care planning documents. The perceived competence we're referring to here is closely tied to the notion of self-efficacy in health behaviors, which has been shown to be an important determinant of health behaviors, healthcare use, and related outcomes. We agree that the statement in question could be more clearly articulated. Thus, we revised the sentence as follow: 

"Finally, in the context of end-of-life decision-making and planning, an individual's perception of their own competence in understanding, discussing, and making informed decisions about end-of-life care can significantly influence their motivation to engage in these important activities." 

Journal Requirements: 

Thank you for your guidance on PLOS ONE's style requirements. We have carefully reviewed these instructions and applied them throughout the manuscript. We have also paid attention to the file naming protocols. We trust that the revised version of the manuscript now aligns with the journal's formatting and style guidelines. However, we remain open to any further feedback or specific areas that may require additional refinement.

2. We note that the grant information you provided in the 'Funding Information' and 'Financial Disclosure' sections do not match. When you resubmit, please ensure that you provide the correct grant numbers for the awards you received for your study in the 'Funding Information' section.

Thank you for pointing out the inconsistency in the grant information provided. We have cross-checked and corrected the grant numbers in the 'Funding Information' section upon resubmission. We appreciate your attention to detail and apologize for any confusion caused.

3. Thank you for stating the following financial disclosure: "SNSF funding for the end-of-life project. Healthy Ageing in the Face of Death: Preferences, Communication, Knowledge, and Behaviors Regarding End of Life and End-of-life Planning Among Older Adults in Switzerland (grant number: 10001C_188836)." Please state what role the funders took in the study. If the funders had no role, please state: "The funders had no role in study design, data collection and analysis, decision to publish, or preparation of the manuscript." If this statement is not correct you must amend it as needed. Please include this amended Role of Funder statement in your cover letter; we will change the online submission form on your behalf.

Thank you for your comment on the role of the funders in our study. I appreciate your detailed guidance on this aspect. We can confirm that "The funders had no role in study design, data collection and analysis, decision to publish, or preparation of the manuscript." This statement accurately represents the funders' involvement in our study. 

4. Thank you for stating the following in the Acknowledgments Section of your manuscript: "The SHARE data collection has been funded by the European Commission, DG RTD through FP5 (QLK6-CT-2001-00360), FP6 (SHARE-I3: RII-CT-2006- 062193, COMPARE: CIT5-CT-2005-028857, SHARELIFE: CIT4-CT-2006-028812), FP7 (SHAREPREP: GA N°211909, SHARE-LEAP: GA N°227822, SHAREM4: GA N°261982, DASISH: GAN°283646) and Horizon 2020 (SHARE-DEV3: GA N°676536, SHARE-COHESION: GAN°870628, SERISS: GA N°654221, SSHOC: GA N°823782, SHARE-COVID19: GA N°101015924) and by DG Employment, Social Affairs & Inclusion through VS 2015/0195, VS 2016/0135, VS 2018/0285, VS 2019/0332, and VS 2020/0313. Additional funding from the German Ministry of Education and Research, the Max Planck Society for the Advancement of Science, the U.S. National Institute on Aging (U01_AG09740-13S2, P01_AG005842, P01_AG08291, P30_AG12815, R21_AG025169, Y1-AG-4553-01, IAG_BSR06-11, OGHA_04-064, HHSN271201300071C, RAG052527A) and from various national funding sources is gratefully acknowledged (see https://share-eric.eu)." 

We note that you have provided funding information that is not currently declared in your Funding Statement. However, funding information should not appear in the Acknowledgments section or other areas of your manuscript. We will only publish funding information present in the Funding Statement section of the online submission form. Please remove any funding-related text from the manuscript and let us know how you would like to update your Funding Statement. Currently, your Funding Statement reads as follows: "SNSF funding for the end-of-life project. Healthy Ageing in the Face of Death: Preferences, Communication, Knowledge, and Behaviors Regarding End of Life and End-of-life Planning Among Older Adults in Switzerland (grant number: 10001C_188836)." Please include your amended statements within your cover letter; we will change the online submission form on your behalf.

Thank you for bringing this discrepancy to attention. The Funding Statement accurately reflects the financial support received for our study - "Healthy Ageing in the Face of Death: Preferences, Communication, Knowledge, and Behaviors Regarding End of Life and End-of-life Planning Among Older Adults in Switzerland". However, we would like to clarify that the funding mentioned in the Acknowledgments Section pertains to the Survey of Health, Ageing and Retirement in Europe (SHARE) data collection, which was crucial for our research but was not funded by our specific project grant. The data collection process was independently funded, and the data was later made accessible for researchers, like us, to conduct secondary analysis. In compliance with the SHARE Conditions of Use, we are required to acknowledge their data collection funding sources in our publication. We understand that this is a unique situation and appreciate your understanding. Here is the link for more information on SHARE's citation requirements: https://share-eric.eu/data/data-access/citation-requirements. We hope this clarifies any confusion, and we are open to further suggestions to ensure we meet the journal's requirements.

5. We note that you have indicated that data from this study are available upon request. PLOS only allows data to be available upon request if there are legal or ethical restrictions on sharing data publicly. For more information on unacceptable data access restrictions, please see http://journals.plos.org/plosone/s/data-availability#loc-unacceptable-data-access-restrictions. In your revised cover letter, please address the following prompts: a) If there are ethical or legal restrictions on sharing a de-identified data set, please explain them in detail (e.g., data contain potentially sensitive information, data are owned by a third-party organization, etc.) and who has imposed them (e.g., an ethics committee). Please also provide contact information for a data access committee, ethics committee, or other institutional body to which data requests may be sent. b) If there are no restrictions, please upload the minimal anonymized data set necessary to replicate your study findings as either Supporting Information files or to a stable, public repository and provide us with the relevant URLs, DOIs, or accession numbers. For a list of acceptable repositories, please seehttp://journals.plos.org/plosone/s/data-availability#loc-recommended-repositories. We will update your Data Availability statement on your behalf to reflect the information you provide.

Thank you for your comments on data availability. We would like to clarify that the data used in this study is not directly managed by us but is part of SHARE. SHARE holds the data and has its own established Conditions of Use, which we adhered to in accessing the data for our study. As per SHARE's policy, access to data is granted to researchers subject to certain conditions designed to protect the anonymity and confidentiality of the survey respondents. These conditions include not sharing the data with others and using it only for the agreed research purpose. Researchers who are interested in accessing the data used in our study can apply to SHARE by following their data access procedure and agreeing to their Conditions of Use. Here is the link for more information: https://share-eric.eu/data/data-access/conditions-of-use. We hope this explanation clarifies our stance on data availability and we appreciate your understanding in this matter.

Thank you for your attention to the details regarding the placement of the ethics statement. We have made the necessary revisions and have now included the ethics statement exclusively in the Methods section of the manuscript:

"Our study received ethical approval from the Ethics Committee of the Canton of Vaud, Switzerland, in March 2014, under the number 66/14." 

We trust this satisfies the journal's requirements.

Thank you for your instructions regarding the Supporting Information files. We have adhered to these and have now included appropriate captions for these files at the end of our manuscript. In addition, we have ensured that any in-text citations are correctly matched.

---

## [Decision Letter · Decision Letter 1]

19 Sep 2023

Development and validation of a subjective end-of-life health literacy scale

PONE-D-23-11080R1

Dear Dr. Meier,

We’re pleased to inform you that your manuscript has been judged scientifically suitable for publication and will be formally accepted for publication once it meets all outstanding technical requirements.

I received one review from a reviewer who had dealt with your manuscript in the initial version, who recommended acceptance. I was unable to obtain a review from the other reviewer. Thus, I have reviewed the revision myself, as second reviewer, and I concur with the decision to accept. I feel you have clearly addressed the primary issues laid out in the reviews and your paper is now stronger and clearer.

Kind regards,

Stefano Occhipinti

Academic Editor

PLOS ONE

Additional Editor Comments (optional):

Reviewers' comments:

Reviewer's Responses to Questions

**Comments to the Author**

1. If the authors have adequately addressed your comments raised in a previous round of review and you feel that this manuscript is now acceptable for publication, you may indicate that here to bypass the “Comments to the Author” section, enter your conflict of interest statement in the “Confidential to Editor” section, and submit your "Accept" recommendation.

Reviewer #2: All comments have been addressed

2. Is the manuscript technically sound, and do the data support the conclusions?

Reviewer #2: (No Response)

3. Has the statistical analysis been performed appropriately and rigorously? 

Reviewer #2: (No Response)

4. Have the authors made all data underlying the findings in their manuscript fully available?

Reviewer #2: (No Response)

5. Is the manuscript presented in an intelligible fashion and written in standard English?

Reviewer #2: (No Response)

6. Review Comments to the Author

Reviewer #2: (No Response)

7. PLOS authors have the option to publish the peer review history of their article (what does this mean?). If published, this will include your full peer review and any attached files.

Reviewer #2: No

---

## [Editor Report · Acceptance letter]

5 Oct 2023

PONE-D-23-11080R1 

*Development and validation of a subjective end-of-life health literacy scale*

Dear Dr. Meier:

I'm pleased to inform you that your manuscript has been deemed suitable for publication in PLOS ONE. Congratulations! Your manuscript is now with our production department. 

Kind regards, 

on behalf of

Prof. Stefano Occhipinti 

Academic Editor

PLOS ONE